# Nanotechnology Innovations in Pediatric Cardiology and Cardiovascular Medicine: A Comprehensive Review

**DOI:** 10.3390/biomedicines12010185

**Published:** 2024-01-15

**Authors:** Stefana Maria Moisa, Alexandru Burlacu, Lacramioara Ionela Butnariu, Corina Maria Vasile, Crischentian Brinza, Elena Lia Spoiala, Alexandra Maștaleru, Maria Magdalena Leon, Solange Tamara Rosu, Radu Vatasescu, Eliza Elena Cinteză

**Affiliations:** 1Department of Pediatrics, Faculty of Medicine, “Grigore T. Popa” University of Medicine and Pharmacy, 700115 Iasi, Romaniaelena-lia.spoiala@umfiasi.ro (E.L.S.); 2“Sfanta Maria” Clinical Emergency Hospital for Children, 700309 Iasi, Romaniasolange.rosu@umfiasi.ro (S.T.R.); 3Faculty of Medicine, “Grigore T. Popa” University of Medicine and Pharmacy, 700115 Iasi, Romania; 4Institute of Cardiovascular Diseases “Prof. Dr. George I.M. Georgescu”, 700503 Iasi, Romania; 5Department of Medical Genetics, Faculty of Medicine, “Grigore T. Popa” University of Medicine and Pharmacy, 700115 Iasi, Romania; 6Pediatric and Adult Congenital Cardiology Department, Centre Hospitalier Universitaire de Bordeaux, 33000 Bordeaux, France; corina-maria.vasile@chu-bordeauz.fr; 7Department of Medical Specialties I, “Grigore T. Popa” University of Medicine and Pharmacy, 700115 Iasi, Romania; alexandra.mastaleru@gmail.com (A.M.);; 8Clinical Rehabilitation Hospital, 700661 Iasi, Romania; 9Department of Nursing, Faculty of Medicine, “Grigore T. Popa” University of Medicine and Pharmacy, 700115 Iasi, Romania; 10Cardio-Thoracic Department, “Carol Davila” University of Medicine and Pharmacy, 020021 Bucharest, Romania; 11Clinical Emergency Hospital, 050098 Bucharest, Romania; 12Department of Pediatrics, “Carol Davila” University of Medicine and Pharmacy, 020021 Bucharest, Romania; eliza.cinteza@umfcd.ro; 13Department of Pediatric Cardiology, “Marie Curie” Emergency Children’s Hospital, 041451 Bucharest, Romania

**Keywords:** nanotechnology, cardiology, pediatric cardiology

## Abstract

(1) Background: Nanomedicine, incorporating various nanoparticles and nanomaterials, offers significant potential in medical practice. Its clinical adoption, however, faces challenges like safety concerns, regulatory hurdles, and biocompatibility issues. Despite these, recent advancements have led to the approval of many nanotechnology-based products, including those for pediatric use. (2) Methods: Our approach included reviewing clinical, preclinical, and animal studies, as well as literature reviews from the past two decades and ongoing trials. (3) Results: Nanotechnology has introduced innovative solutions in cardiovascular care, particularly in managing myocardial ischemia. Key developments include drug-eluting stents, nitric oxide-releasing coatings, and the use of magnetic nanoparticles in cardiomyocyte transplantation. These advancements are pivotal for early detection and treatment. In cardiovascular imaging, nanotechnology enables noninvasive assessments. In pediatric cardiology, it holds promise in assisting the development of biological conduits, synthetic valves, and bioartificial grafts for congenital heart defects, and offers new treatments for conditions like dilated cardiomyopathy and pulmonary hypertension. (4) Conclusions: Nanomedicine presents groundbreaking solutions for cardiovascular diseases in both adults and children. It has the potential to transform cardiac care, from enhancing myocardial ischemia treatment and imaging techniques to addressing congenital heart issues. Further research and guideline development are crucial for optimizing its clinical application and revolutionizing patient care.

## 1. Introduction

Nanotechnologies have significantly enhanced patient care by advancing the fields of biotechnology, medicine, and pharmaceuticals. These technologies have streamlined various healthcare processes, encompassing diagnosis, therapeutic interventions, and ongoing patient monitoring. An ongoing effort is being made to innovate and develop new nanomaterials to improve disease diagnosis and treatments. This innovation seeks to ensure that medical practices become more personalized, cost-effective, and safe, enhancing their potency, accuracy, and duration [1,2]. The future of nanotechnology hinges on selecting appropriate nanomaterials while mitigating potential adverse effects. Like all medical products, nano-based products must undergo rigorous risk evaluations before clinical and commercial approval to minimize potential risks to human health and the environment. A comprehensive life cycle assessment is crucial to determine their usage’s long-term sustainability and safety [3].

Nanotechnology constitutes a relatively new area of science, encompassing the investigation of nanoparticles and nanomaterials with dimensions less than 100 nm, which have unique advantages such as increased surface area, tunable physicochemical properties, and the ability to encapsulate various therapeutic agents [4]. The European Medicines Agency defines nanomedicine as implementing nanotechnology in different medical domains, from prevention and diagnosis to targeted therapeutic interventions [5].

Several types of nanoparticles are available, and some of them have been explored in human subjects, including polymer-based (e.g., dendrimers, micelles, nanogels, drug conjugates), lipid-based (liposomes, exosomes, solid lipid nanoparticles), non-polymeric (e.g., nanotubes, nanodiamonds, metallic nanoparticles) and nanocrystalline nanoparticles [6,7,8] (Figure 1).

Lipid-based nanoparticles, such as liposomes and lipid nanoparticles, have gained prominence for drug delivery in cardiovascular diseases. Liposomes can encapsulate hydrophobic and hydrophilic drugs, providing a versatile platform for controlled release. These nanoparticles enhance drug stability, improve pharmacokinetics, and reduce systemic toxicity [9]. Polymeric nanoparticles, including dendrimers and micelles, offer customizable structures for cardiovascular drug delivery [10]. Dendrimers exhibit a highly branched architecture, enabling precise control over drug release kinetics [11]. Micelles, composed of amphiphilic polymers, enhance drug solubility and circulation time, leading to improved therapeutic outcomes [9,12]. Inorganic nanoparticles, such as gold and silica nanoparticles, are utilized for their unique optical and magnetic properties. Gold nanoparticles serve as contrast agents in cardiovascular imaging, enabling precise detection of lesions and monitoring of treatment responses [13]. Silica nanoparticles, with their porous structure, facilitate controlled drug release and targeted therapy [14]. Magnetic nanoparticles, often based on iron oxide, provide a platform for targeted drug delivery and imaging guided by external magnetic fields. These nanoparticles enable site-specific drug release, enhancing drug concentration at the desired cardiovascular site while minimizing systemic exposure [15]. Carbon-based nanoparticles, including carbon nanotubes and graphene, have shown potential in cardiovascular applications [16]. Their high surface area and unique electronic properties make them suitable for drug delivery and imaging. However, concerns regarding biocompatibility and long-term safety necessitate further research [16].

Although nanoparticles have proved efficient in various biomedical areas, extensive clinical applicability is relatively limited. Some key factors that might perpetuate clinical translation reluctance are safety and toxicity concerns, unclear definitions and physical properties of nanoparticles, cost–efficiency ratios, and ambiguous regulatory guidelines [7]. Also, other concerns regarding the biocompatibility of nanoparticles, pharmacokinetic and pharmacodynamic proprieties, and manufacturing process issues should be addressed when implementing nanomedicine in clinical practice [17].

Nanoparticle safety concerns are driven mainly by physical properties, such as size and shape, solubility, agglomeration state, and surface area [18]. Concerning the link between size and safety, one study documented that only small gold nanoparticles (1.4 nm) were toxic, in comparison to bigger nanoparticles (15 nm) which were not [19]. Nevertheless, technological progress has continuously improved the safety profile of nanoparticles in the last few years. Recently, Sani et al. emphasized that, even at present, small nanoparticles remain the most controversial regarding safety [20]. However, nanotechnology implementation in clinical practice is trending upwards, as many products have been approved for use in adults and even in children [18].

Severe acute respiratory syndrome coronavirus 2 (SARS-CoV-2) vaccines (Pfizer-BioNTech, Moderna, Cambridge, MA, USA) represent one of the most extensive utilizations of nanoparticles in humans [6]. Nevertheless, nanoparticles encompass far more than SARS-CoV-2 vaccines. In recent decades, the Food and Drug Administration (FDA) has approved an impressive arsenal of nanoparticles for different clinical applications [3]. Consequently, nanoparticles have the potential to be used to manage various types of cancer, infections, transthyretin-mediated amyloidosis, multiple sclerosis, anemia (including patients with chronic kidney disease), and autoimmune, inflammatory, and metabolic disorders [1].

Moreover, the FDA has approved some nanoparticles for administration in pediatric patients. In this regard, sodium ferric gluconate can be used in children ≥6 years old with end-stage kidney disease (ESKD) on hemodialysis for anemia correction due to iron deficiency [6,21]. Octocog alfa is a lipid-based nanoparticle loaded with human coagulation factor VIII and has been approved for hemophilia A therapy irrespective of patients’ age [6,22]. Elixophyllin is a crystalline nanoparticle containing theophylline and has been approved for chronic asthma treatment and other conditions associated with airflow limitation [3,20,23]. Also, nanoparticles could be used for cancer therapy in children. Mepact, a lipid-based nanoparticle loaded with mifamurtide, is indicated in patients with osteosarcoma in the postoperative setting, combined with other adjunctive therapies [6,24].

This review seeks to critically assess the current advancements in nanotechnology within cardiology and pediatric cardiology. We aim to investigate the benefits and limitations of nanomedicine in treating cardiovascular disorders, thereby setting the stage for future research and the development of clinical guidelines in this rapidly evolving sector.

## 2. Methodology

We conducted an extensive literature search across multiple databases, including PubMed, Embase, and ScienceDirect, employing the following search terms: “nanotechnology”, “cardiology”, “pediatric cardiology”, “children”, “cardiovascular”, and “pediatric patients”. The inclusion criteria encompassed the following key elements: articles available as full texts, studies published within the last 20 years (from 2003 onwards), clearly indicating the type of nanotechnology utilized and its relevance to either cardiology or pediatric cardiology, as well as articles presented in English. We considered various studies, including clinical, preclinical, animal research, literature reviews, and ongoing clinical trials. Additionally, we examined the reference lists of the included articles to identify further relevant studies. Exclusion criteria encompassed single case reports, letters to the editor, and studies available only in abstract form.

Our search revealed 74 potential articles. After removing duplicates and articles not meeting the inclusion criteria, 58 records were left for further analysis. The selection process involved a two-step evaluation conducted independently by two authors. In the first step, titles and abstracts were screened to determine eligibility. Subsequently, all studies meeting the inclusion criteria progressed to the second step, where the full texts of articles were thoroughly examined.

## 3. Results

This review included 33 research papers in the final analysis. These articles explored various nanotechnological approaches aimed at addressing challenges such as heart ischemia, pulmonary hypertension, and the enhancement of medical imaging techniques. As can be seen in the tables below, the included studies were performed on in vitro models (Table 1), animal models (Table 2), or humans (Table 3).

Notably, within this analysis, approximately a quarter of studies prioritized safety assessments of the proposed methods involving nanotechnologies, while the majority reported favorable efficacy outcomes.

### 3.1. Nanotechnology Imaging

Unstable atherosclerosis plaques are rich in macrophages. A novel imaging nanoparticle method involves tracing the metabolic activity of these macrophages by glucose analog uptake, specifically fluorine-labeled 2-deoxy-D-glucose (FDG) uptake. Then, positron emission tomography is employed to describe the state of the suspected atherosclerotic artery [56]. Unstable plaques are additionally characterized by a significant population of apoptotic cells, detectable via the utilization of annexin labeled with radionuclides [49,58]. The next step in vascular imaging is to detect the dysfunctional endothelium before the plaque is formed. This can be achieved by targeting VCAM-1 [51], E-selectin [37] or P-selectin [51,59].

Imaging can also be extrapolated to the electrical activity of the heart. In this regard, Shimozawa and colleagues [38,60] have shown that cardiomyocyte electrical connectivity encompasses not only gap junctions and intercalated disks but also implicates mechanosensitive cardiac sodium channels. Employing real-time confocal imaging, they demonstrated that intercalated disks were flexible and varied in length during action potential propagation from one cardiomyocyte to another, leading to mechanical stress that activates cardiac sodium channels [60].

Nanomaterial-based noninvasive molecular imaging employs magnetic resonance imaging, optical imaging, nuclear scintigraphy, photoacoustic imaging, single photon emission computed tomography, optical coherence tomography/infrared luminescence, and X-ray-excited luminescence. These methods employ iron oxide, gadolinium, polylactide polycarboxybetaine or metal oxide-peptide amphiphile micelles as effective nanoparticles for imaging atherosclerotic plaque thrombosis. P selectin is overexpressed in inflamed endothelial cells. Nanoparticles composed of fucoidan and a thermolysin-hydrolyzed protamine peptide can be combined with magnetic resonance imaging to identify these endothelial cells. Dual-contrast iron oxide nanoparticles can be used as MRI contrast agents. Vulnerable atherosclerotic plaques can easily be identified using 18F-fluorodeoxyglucose (18F-FDG)-based positron emission tomography (PET), but this method lacks specificity [61].

### 3.2. Myocardial Ischemia

Heart failure is a preeminent global healthcare concern, ranking as the leading cause of mortality worldwide. Its etiology encompasses various factors, including cardiomyopathies, congenital cardiac anomalies, hypertension, valvular lesions, myocarditis, arrhythmias, drug-induced effects, neuromuscular disorders, and ischemic heart disease. Among adults, ischemic heart disease overwhelmingly predominates as the main precipitating factor of heart failure. Furthermore, ischemic heart disease imposes a substantial burden on public health, significantly increasing morbidity and mortality (even in the absence of heart failure).

The early detection of myocardial ischemia is crucial in clinical practice. Established biomarkers such as cardiac troponins, myoglobin, and creatine kinase MB are employed to identify ischemic events within the bloodstream. However, these biomarkers are characterized by initially low peripheral blood concentrations. A promising step forward lies in the integration of biosensors and nanotechnology. Several laboratory methodologies can be enlisted for this purpose, including electrochemistry, chemiluminescence, electrochemiluminescence, photoelectrochemistry, fluorescent assays, cyclic voltammetry, electrochemical impedance spectroscopy, Lateral Flow Immunosensors, Lateral Flow assays, and differential pulse voltammetry [61].

Coronary atherosclerosis, the underlying pathology responsible for myocardial ischemia, may be effectively addressed through nanotechnologies. Several release systems (polyethylene glycol (PEG), poly lactic-co-glycolic acid (PLGA), or poly(d-lactic acid) [43] are nowadays available to deliver anti-inflammatory molecules such as Ac2-26 peptide [47] and interleukin-10 cytokine [38] to the atherosclerotic plaque. Foam cell formation can also be inhibited using macromolecules by suppressing macrophage scavenger receptors using sugar-based amphiphilic macromolecules (Figure 2) [55]. However, several aspects still need to be addressed, such as the potential proinflammatory effect of such technologies via fibrinogen conformational change induction [62].

Another way to attempt reperfusion is to transplant nanomaterials (aligned nanofibrillar scaffolds) impregnated with stromal vascular fraction cells. This method proved superior to phosphate buffer saline administration in a mouse model [58]. Furthermore, another alternative for reperfusion exists: vessel grafts with anthracene-grafted styrene-block-butadiene block-styrene, which proved to be a promising vascular tissue-engineering alternative in a recent study [25].

When using conventional stents for coronary angioplasty, there is an early risk of thrombosis and a delayed risk of restenosis. These risks could be reduced by inserting nitric oxide-producing stents [63]. This can be achieved using nitric oxide-releasing or nitric oxide-generating coatings for stents. Drug-eluting stents represent a notable advancement over bare metal stents for maintaining vessel patency. However, they remain susceptible to late thrombosis as a result of drug depletion [32]. Nitric oxide donors serve as vehicles for the controlled delivery of nitric oxide, releasing it selectively in targeted regions for a finite duration of time [64]. Nitric oxide donors include S-nitroso-N-acetylpenicillamine (SNAP) and N-diazeniumdiolates (NONOates) [64].

Gene therapy has been proposed as a solution to the limited nitric oxide donation time problem. This can be achieved either by transfection of autologous adipose-derived stem cells (ASCs) or differentiated endothelial-like cells [48], or by using iNOS-encoding adeno-associated virus serotype 2 (AAV2) vectors [28].

Myocardial repair is also thought to be achievable using nanotechnology in the future. One study [25] showed optimistic results by using human induced pluripotent stem cell-derived cardiomyocytes (hiPSC-CMs) cultured in a three-dimensional micro-environment, conjugated with an aligned polycaprolactone (PCL)-gelatin coaxial nanofiber patch fabricated using electrospinning. An alternative strategy involves tagging fresh cardiomyocytes with magnetic nanoparticles and immobilizing them by applying a localized magnetic field [45].

Nevertheless, cell therapy is not exempt from associated risks, and its potential arrhythmic impact is under thorough investigation [45]. One alternative is synthesizing patient-specific cardiomyocytes from stem cells [65], but they might be subject to immune rejection [66]. Considering the substantial presence of fibrin within infarcted myocardial tissue, it is postulated to constitute an auspicious candidate for precisely targeting patient-specific therapeutic cardiomyocytes or alternative modalities for cardiac regeneration therapy, provided such interventions are administered promptly. Nonetheless, fibrin formation is not restricted exclusively to the infarcted region, presenting a significant drawback in this approach. More selective molecular targets include myosin, angiotensin II type 1 receptor, and phosphatidylserine [67].

While myocardial repair following an acute ischemic episode holds utmost significance, it is equally imperative to detect the onset of ischemia promptly. One way to achieve this is the detection of cardiac troponin I using a multi-functional DNA on Au nanocrystal-modified indium-tin-oxide substrate using an electrochemical method [27].

### 3.3. Myocarditis

Accurate diagnosis is pivotal for effective myocarditis management. Nanoparticles, including superparamagnetic iron oxide nanoparticles (SPIONs) and quantum dots, have demonstrated considerable potential as contrast agents for imaging modalities such as magnetic resonance imaging (MRI) and fluorescence imaging [68]. These nanomaterials enhance sensitivity and specificity, allowing for early detection and precise localization of inflammatory lesions in the myocardium [68].

Nanoparticles designed for targeted drug delivery present a paradigm shift in myocarditis therapeutics. Functionalized nanocarriers can deliver anti-inflammatory agents directly to the inflamed myocardium, minimizing systemic side effects. Toita et al. proposed a bioinspired anti-inflammatory nanomedicine conjugated with protein G that target proinflammatory cells, such as macrophages [69]. This targeted approach is promising for improving treatment efficacy and reducing adverse effects on healthy tissues.

In animal models, Li et al. showed that using nano-alpha-linolenic acid in mice with viral-induced myocarditis improved their survival rate in a dose-dependent manner, but further research in humans is needed to replicate and extend these results [70].

### 3.4. Aneurysms

Aneurysms are a rare but serious condition requiring innovative approaches. Nanotechnology, operating at the nanoscale, offers a unique tool for precise diagnosis and treatment. Currently, most results are based on animal models, and few clinical trials have been conducted. In rodent studies, targeted nanoparticles showed promising results in abdominal aortic aneurysms [71]. This represents a groundbreaking approach to enhancing the efficacy of pharmacological interventions while minimizing systemic side effects. Functionalized nanocarriers can deliver therapeutic agents directly to the aneurysm site, improving drug concentration and reducing off-target effects. This approach holds great promise for pediatric patients, where precise drug delivery is crucial for minimizing adverse effects on developing organs. However, surgical and endovascular treatment remains the preferred strategy to address these cases [72].

Collaborative efforts between researchers and clinicians are essential for establishing guidelines and standards for nanotechnological interventions in pediatric populations. However, nanotechnology holds immense potential for revolutionizing children’s diagnosis and the treatment of aneurysms. Integrating nanomaterials into imaging, monitoring, drug delivery, and surgical guidance opens new avenues for personalized and minimally invasive interventions. Further research is required to address challenges related to biocompatibility, safety, and ethical considerations, paving the way to translating these exciting advancements into clinical practice.

### 3.5. Dilated Cardiomyopathies

Patients with dilated cardiomyopathy can benefit from pacing that bypasses scar tissue, which can be accomplished by using thread-like carbon nanotube fibers with high electrical conductivity [73]. Attempts have also been made to treat cardiomyopathies by inducing the formation of new mature cardiomyocytes [53], but the results were unsatisfactory. In some studies, attempts were made to inject several types of bone marrow-derived cells into dilated cardiomyopathy hearts, but severe side effects such as arrhythmia or tumors arose [74,75,76,77]. Other authors attempted to transform fibroblasts into contractile cardiomyocytes [75] by using cardiac-specific transcription factors or mitochondrial RNA to induce cardiomyocyte regeneration and proliferation [40]. Growth factors have been linked to nanoparticles and delivered to the heart to induce neovascularization [57], but this method is not yet used in clinical practice.

### 3.6. Pulmonary Hypertension

Pulmonary hypertension in children poses a significant challenge due to its complex pathophysiology and limited therapeutic options. Nanotechnology has emerged as a promising frontier for innovative and targeted interventions. In mice, Marulanda et al. proposed an intravenous delivery system for lung-targeted nanofibers, aiming to decrease pulmonary hypertension [76]. Their results showed promising perspectives, especially for specific proteins overexpressed in this condition, such as the receptor for advanced glycation end-products (REGE) [76].

Endothelial dysfunction and vasoconstriction are hallmarks of pediatric pulmonary hypertension. Nanoparticles can be engineered to release vasodilatory agents specifically in the pulmonary vasculature. Recently, Elbardisy et al. reported that aerosolized dilutable nanoemulsions of tadalafil showed a favorable safety profile in both in vitro and in vivo toxicity studies [77]. This targeted approach holds promise for improving blood flow and alleviating symptoms in children with pulmonary hypertension [77].

Children with pulmonary hypertension can benefit from bosentan nanoparticle treatment, which increases the bioavailability of the molecule by several times [29]. Akagi et al. [42] used a rat model to study beraprost nanoparticles for pulmonary hypertension treatment, with favorable results. Pulmonary arterial smooth muscle cell proliferation was effectively inhibited by using imatinib nanoparticles [44]. Promising results were also obtained using fasudil [50], nuclear factor kappa B [53], or pitvastatin [52] nanoparticles.

While nanotechnology holds great promise in pediatric pulmonary hypertension management, challenges such as regulatory approval, scalability, and long-term safety must be addressed. The safe integration of nanotechnology into pediatric treatment requires rigorous evaluation of biocompatibility and potential long-term effects.

Furthermore, developing biodegradable nanocarriers addresses concerns about long-term retention and potential toxicity [78].

### 3.7. Vascular Tissue Engineering

Pediatric cardiology would highly benefit from manufacturing biological conduits; however, nowadays, science faces serious challenges in modeling compact structures. We are still in the era of monolayer cell groups, but a biological conduit or synthetic valves that would have the capability of growing at the same pace as the host body would signify a great leap forward. These conduits would have to be lined with endothelial cells. Golden et al. [35] have taken steps toward obtaining a network of narrow (6 μm) channels. The next step was seeding endothelial cells in this network. Moving on to three-dimensional structures, they met geometrical issues: corners of such structures impair blood flow and result in asymmetrical parietal stress. Their work was carried on by Borenstein et al. [47], who obtained semicircular cross-sections and improved blood flow at bifurcations. Chrobak et al. [36] used a needle-collagen technique to obtain a tube network lined with endothelial cells that could be perfused with insignificant leakage.

Skardal et al. [34] attempted to manufacture a bioartificial vessel-like graft by culturing microfilaments that fused together. Suri et al. [31] used hyaluronic acid and a digital mirror device to fabricate layered structures, while Duan et al. [30] used an ultraviolet-led module to manufacture artificial valves with a similar geometry to biological valves.

The application of nanomedicine in coronary artery bypass grafting (CABG) represents a significant stride in enhancing the efficacy and safety of this crucial cardiac procedure. Nanotechnology in CABG focuses on improving graft materials and enhancing post-surgical healing and integration. Nanoscale modifications of graft surfaces can promote better endothelialization, reducing the risk of thrombosis and improving long-term patency rates [79]. Furthermore, nanomaterials can be engineered to deliver therapeutic agents directly to the graft site, aiding in the suppression of local inflammation and the prevention of neointimal hyperplasia, a common cause of graft failure. This targeted delivery system allows for a more efficient use of drugs, potentially reducing systemic side effects. Additionally, nanoscale imaging agents can provide surgeons with enhanced visualization during procedures, ensuring more precise graft placement. As research advances, the integration of nanomedicine in CABG is poised to revolutionize the approach to coronary artery disease treatment, offering patients improved outcomes and quicker recoveries [80].

Nanomedical processes in cardiac repair, particularly for nerve regeneration, involve advanced materials and techniques that mimic the heart’s extracellular matrix. These nanoscale innovations, such as electrically conductive nanomaterials and nanofibers, provide crucial support for the growth and healing of cardiac nerves. They can also deliver growth factors and other regenerative molecules directly to the damaged cardiac tissue. This integration of nanotechnology with regenerative medicine, like stem cell therapy, enhances the effectiveness of cardiac repair, aiming to restore the heart’s normal function and electrical connectivity after injury. As this field is still developing, ongoing research is crucial for realizing the full potential of these nanomedical strategies in heart health [81].

### 3.8. Congenital Heart Disease

Congenital heart diseases are another field where coated nanotechnology devices can be used. Percutaneous closures offer several advantages over surgical thoracotomies, such as avoiding cardiopulmonary bypass, avoiding sternotomies, and potentially lowering postgraduate complications [78,82].

Closing different types of cardiac malformation (atrial septal defect, ventricular septal defect, patent ductus arteriosus, patent foramen ovale, etc.) may require cardiac devices. Most modern cardiac devices that undergo percutaneous closure are made of nitinol, an alloy of nickel and titanium. Nickel has been shown to leak from nitinol-containing occluders in previous studies. The mean serum nickel levels after Amplatzer device implantation rose significantly within 24 h of implantation and remained elevated until 1 month after implantation before gradually decreasing to baseline levels. In patients with ASD, Burian et al. also found that nickel levels rose significantly after implantation of a nickel device in serum and urine; nickel levels in serum increased up to fivefold after the 6-week post-closure period, and nickel levels returned to baseline levels after 4 to 6 months. After implanting the Amplatzer occluder, serum nickel levels rose dramatically, peaking at three months after implantation, and slowly dropping back to baseline during follow-up [83,84,85].

There is a risk of nickel allergy associated with the release of the nickel with time in some cases of these devices. The transcatheter closure of cardiac shunts has been associated with systemic adverse effects of nickel allergy, including pericarditis and migraine headaches. It may be necessary to surgically explant the device in patients with nickel allergy and severe refractory symptoms [86,87]. To prevent nickel allergy, some companies have developed nanotechnologies to coat the devices with nanoparticles of platinum, bioceramic, or polymers (parylene) to prevent nitinol externalization [82,88,89,90].

Recent advancements in cardiac repair have shone a spotlight on the potential of extracellular matrix (ECM)-based cardiac patches [91]. These patches, engineered to mimic the natural ECM of the heart, have emerged as a promising therapeutic strategy for repairing damaged myocardial tissue. The ECM, a complex network of proteins and polysaccharides, provides essential structural and biochemical support to cells. By replicating this environment, ECM-based patches facilitate the integration and proliferation of cardiac cells, thereby aiding in the regeneration of injured myocardial regions [91].

A key attribute of these cardiac patches is their ability to provide a conducive microenvironment for the recruitment and maturation of cardiomyocytes and vascular cells, which is essential for restoring heart function. This is particularly significant in conditions like myocardial infarction, where the loss of viable heart tissue leads to compromised cardiac function. The application of ECM-based patches in such scenarios has shown promising results in promoting angiogenesis and myocardial regeneration [92].

Moreover, the biocompatibility and biodegradability of these patches align well with the body’s natural healing processes, minimizing the risk of adverse immune responses. This compatibility is crucial for ensuring the long-term success of cardiac repair and regeneration therapies. Ongoing research is focused on enhancing the mechanical properties and bioactivity of these patches to better mimic the native cardiac tissue, thus improving their therapeutic efficacy [91]. Overall, ECM-based cardiac patches represent a significant leap forward in regenerative medicine. By leveraging the innate properties of the extracellular matrix, they offer a sophisticated approach to cardiac repair, with the potential to significantly improve outcomes for patients with heart diseases [91].

The development of sirolimus or paclitaxel drug-eluting stents (DESs) has revolutionized coronary intervention by reducing intimal growth. In addition to their indication for myocardial ischemia, off-label DESs are used in pediatric cardiology with good results as palliation in hybrid interventions for hypoplastic left heart syndrome or different forms of pulmonary atresia in infants with duct-dependent circulation. Neonates with duct-dependent pulmonary circulation can benefit from ductal stents instead of Blalock–Taussig shunts. DESs are also preferred in children because they maintain a longer duct patency than bare-metal stents [93,94]. However, DESs contain paclitaxel or sirolimus, which may cause concern for neonates regarding their immunosuppressive effects, especially if more than one stent is used or one longer than 22 mm. A study by Sivakumar et al. evaluated the increasing levels of sirolimus after stent implantation in neonates with pulmonary duct-dependent circulation. Immunosuppressive sirolimus levels ranged from 5 to 15 ng/mL, and this was not surpassed whether one or two stents were used [95].

## 4. Conclusions

In this comprehensive narrative review, we have explored the landscape of nanotechnology advancements in cardiovascular medicine and pediatric cardiology. Nanomedicine, encompassing a diverse array of nanoparticles and nanomaterials, has emerged as a promising frontier in medical science. Despite initial reservations and concerns, substantial progress has been made, leading to the approval and utilization of various nanotechnology-based interventions in adult and pediatric patients. Myocardial ischemia, a critical concern in cardiology, has seen innovative approaches involving nanomaterials, including drug-eluting stents, nitric oxide donors, and magnetic nanoparticle-tagged cardiomyocytes. These developments hold immense potential for improving patient outcomes, particularly in the early detection and treatment of ischemic events. Nanotechnology has also revolutionized imaging techniques, enabling the noninvasive assessment of vascular and cardiac structures. Concerning pediatric cardiology, creating biological conduits, synthetic valves, and bioartificial vessel-like grafts represents a future research direction for addressing congenital heart defects. Moreover, nanotechnology-based treatments have shown potential in mitigating conditions such as dilated cardiomyopathy and pulmonary hypertension in children, thereby expanding therapeutic options for this vulnerable population. These findings lay the foundation for continued research and the development of guidelines to optimize the clinical utility of nanomedicine in cardiology.

## Figures and Tables

**Figure 1 biomedicines-12-00185-f001:**
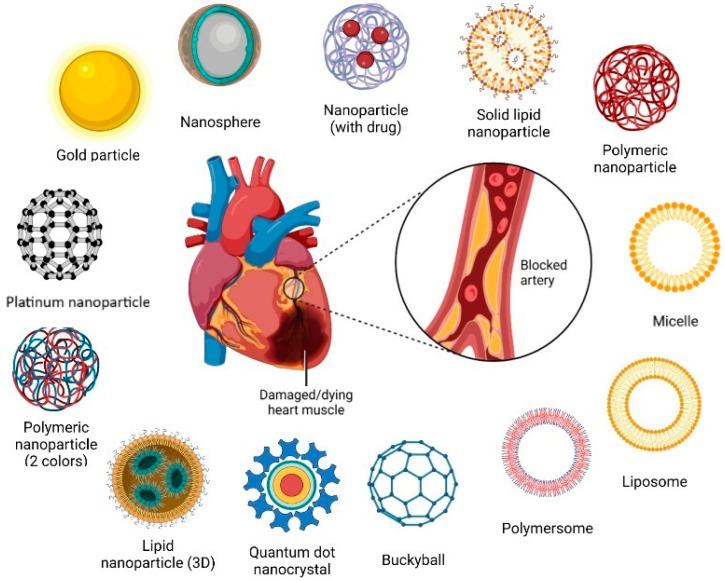
Potential useful nanoparticles in cardiovascular diseases. Created with Biorender.com.

**Figure 2 biomedicines-12-00185-f002:**
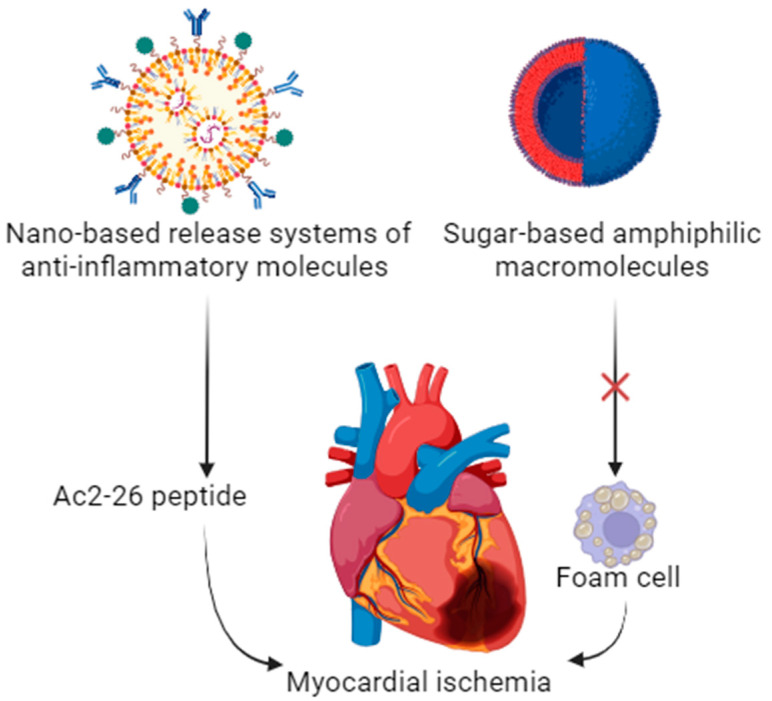
Potential application of nanoparticles in myocardial ischemia. Created with Biorender.com.

**Table 1 biomedicines-12-00185-t001:** In vitro studies analyzing potential roles of nanomedicine in cardiovascular medicine.

Reference	Year	Utility	Type of Nanotechnology	Reported Efficacy	Safety Outcome
Tan et al. [25]	2020	Vascular tissue engineering	Aligned patterned biomimetic elastic membrane	Yes	N/A
Kumar et al. [26]	2020	Myocardial infarction	Aligned polycaprolactone (PCL)-gelatin coaxial nanofiber patch	Yes	N/A
Lee et al. [27]	2019	Cardiac troponin I-based diagnosis systems	Dual-mode cardiac troponin I biosensor comprised of multi-functional DNA on Au nanocrystal	Yes	No
Fishbein et al. [28]	2017	Atherosclerosis	Adeno-associated viral (AAV) vectors	Yes	N/A
Ghasemian et al. [29]	2016	Pulmonary arterial hypertension	Nanosuspension	Yes	No
Duan et al. [30]	2014	Valve replacement	3-D printable formulations of hybrid hydrogels	Yes	No
Suri et al. [31]	2011	Nerve tissue engineering	Hyaluronic acid scaffolds	Yes	No
Ma et al. [32]	2010	Atherosclerosis	Glycogen synthase kinase-3beta inhibitor	Yes	N/A
Borenstein et al. [33]	2010	Vascular tissue engineering	Polystyrene microvascular networks	Yes	No
Skardal et al. [34]	2010	Vascular tissue engineering	Hyaluronan hydrogels crosslinked with tetrahedral polyethylene glycol tetracrylates	Yes	No
Golden et al. [35]	2007	Vascular tissue engineering	Micromolded meshes of gelatin	Yes	No
Chrobak et al. [36]	2006	Vascular tissue engineering	Monolayers of human endothelial cells	Yes	No
Mulder et al. [37]	2004	Imaging diagnosis techniques	Pegylated paramagnetic fluorescently labeled liposomes carrying anti-E-selectin monoclonal antibody	Yes	No

**Table 2 biomedicines-12-00185-t002:** Animal studies analyzing potential roles of nanomedicine in cardiovascular medicine.

Reference	Year	Utility	Type of Nanotechnology	Reported Efficacy	Safety Outcome
Kobirumaki-Shimozawa et al. [38]	2020	Imaging diagnosis techniques	CellMask	Yes	N/A
Hu et al. [39]	2020	Peripheral arterial disease	Human stromal vascular fraction cells on nanofibrillar scaffolds	Yes	Yes
Gao et al. [40]	2019	Heart failure	iR-19a/19bmimics	Yes	No
Kamaly et al. [41]	2016	Atherosclerosis	Interleukin-10 nanotherapeutics developed with a microfluidic chip	Yes	No
Akagi et al. [42]	2016	Pulmonary arterial hypertension	Beraprost-incorporated nanoparticles	Yes	Yes
Fredman et al. [43]	2015	Atherosclerosis	Proresolving peptide Ac2-26	N/A	No
Akagi et al. [44]	2015	Pulmonary arterial hypertension	Imatinib-incorporated nanoparticles	Yes	No
Vandergriff et al. [45]	2014	Myocardial infarction	Human cardiosphere-derived stem cells labeled with FDA-approved ferumoxytol nanoparticles	Yes	Yes
Chong et al. [46]	2014	Myocardial infarction	Human embryonic stem cell-derived cardiomyocytes	No	Yes
Kamaly et al. [47]	2013	Atherosclerosis	Peptide Ac2-26 (annexin A1/lipocortin 1-mimetic peptide)	Yes	No
McIlhenny et al. [48]	2013	Vascular tissue engineering	Autologous ASC transfected with the endothelial nitric oxide synthase (eNOS) gene	Yes	No
De Saint-Hubert et al. [49]	2013	Atherosclerosis	124I-Hypericin	Yes	No
Gupta et al. [50]	2013	Pulmonary arterial hypertension	Liposomal fasudil	Yes	No
Jacobin et al. [51]	2011	Atherosclerosis	Versatile ultrasmall superparamagnetic iron oxide (VUSPIO) particles	Yes	No
Chen et al. [52]	2011	Pulmonary arterial hypertension	Pitvastatin fluorescein isothiocyanate lactide/glycolide copolymer	Yes	No
Kimura et al. [53]	2009	Pulmonary arterial hypertension	Nuclear factor κB -κB decoy oligodeoxynucleotides	Yes	No
McAteer et al. [54]	2008	Atherosclerosis	Microparticles of iron oxide	Yes	No
Chnari et al. [55]	2006	Atherosclerosis	Nanoblocker macromolecules containing mucic acid, lauryl chloride, and poly(ethylene glycol)	Yes	No

**Table 3 biomedicines-12-00185-t003:** Human studies analyzing potential roles of nanomedicine in cardiovascular medicine.

Reference	Year	Utility	Type of Nanotechnology	Reported Efficacy	Safety Outcome
Dweck et al. [56]	2011	Aortic stenosis	18F-Flurodeoxyglucose	Yes	No
Henry et al. [57]	2003	Myocardial infarction	Recombinant human vascular endothelial growth factor protein	Yes	Yes

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
