# Peer review of "Nanotechnology Innovations in Pediatric Cardiology and Cardiovascular Medicine: A Comprehensive Review"

_biomedicines, 2024, doi:10.3390/biomedicines12010185_

Round 1

Reviewer 1 Report

Comments and Suggestions for Authors

It is a well-written review manuscript. 

The reviewer hope the authors can expand some discussions about potential Nanotechnology and gene therapy in heart diseases, especially in Congenital heart disease. Is there possibility to correct some defects in congenital heart disease through nanoparticle-mediated gene threapy ?

Author Response

Dear Editor,

We are grateful to the Editorial Board for considering a revised version of our manuscript biomedicines-2773422 entitled “Nanotechnology Innovations in Pediatric Cardiology and Cardiovascular Medicine: Comprehensive Review” in the Biomedicines Journal..

We would like to thank the reviewers for their valuable time and useful contribution. We much appreciate their input, which helped improve our manuscript.

Also, we look forward to hearing from you regarding our submission. We would be glad to respond to any further questions and comments that you may have.

Please find below a detailed point-by-point reply to the comments made by the Reviewer 1.

Sincerely,

--

Alexandru Burlacu

MD, Ph.D., FESC

Assoc. Professor @ University of Medicine and Pharmacy “Gr. T. Popa” Iasi – ROMANIA

Interventional Cardiologist – Senior Consultant

Department of Interventional Cardiology – Cardiovascular Diseases Institute – Iasi

ORCID: https://orcid.org/0000-0002-3424-1588

Scopus Author ID: 56811734600

On behalf of the Authors

Reviewers’ Comments to Authors:

Reviewer 1:

It is a well-written review manuscript.

Answer 0.

We would like to thank the Reviewer 1 and the Editorial board for all the positive remarks regarding our work.

We are delighted to hear that the Reviewer 1 observed the quality of the manuscript and the in-depth analysis of the subject.

We assure the EIC and the Reviewer 1 that we have read carefully every suggestion from this decision letter and tried our best to improve the quality of the document accordingly.

Q1. The reviewer hope the authors can expand some discussions about potential Nanotechnology and gene therapy in heart diseases, especially in Congenital heart disease. Is there possibility to correct some defects in congenital heart disease through nanoparticle-mediated gene threapy ?

Answer 1:

Thank you for your excellent point!

The reviewer is correct in stating that there is a potential for nanotechnology to be used to correct defects in congenital heart disease. Nanomaterials can be used to deliver therapeutic agents directly to the affected area, which can help to improve the effectiveness of treatment and reduce the risk of side effects. Additionally, nanotechnology can be used to create new biological materials that can be used to repair or replace damaged heart tissue. While the field of nanoparticle-enhanced gene therapy for cardiology has shown promise in preclinical studies, there is a lack of data on the long-term safety and efficacy of this approach in humans. Further research is needed to develop nanoparticles that can effectively deliver genes to the heart while minimizing off-target effects, ensure that the genes delivered by nanoparticles are expressed in the correct cells and at the appropriate level, and assess the long-term safety of nanoparticle-enhanced gene therapy. Following the Reviewer recommendations, we provided additional data in our manuscript, as follows:

“Recent advancements in cardiac repair have spotlighted the potential of extracel-lular matrix (ECM)-based cardiac patches [88]. These patches, engineered to mimic the natural ECM of the heart, have emerged as a promising therapeutic strategy for re-pairing damaged myocardial tissue. The ECM, a complex network of proteins and pol-ysaccharides, provides essential structural and biochemical support to cells. By repli-cating this environment, ECM-based patches facilitate the integration and proliferation of cardiac cells, thereby aiding in the regeneration of injured myocardial regions [88].

A key attribute of these cardiac patches is their ability to provide a conducive mi-croenvironment for the recruitment and maturation of cardiomyocytes and vascular cells, essential for restoring heart function. This is particularly significant in conditions like myocardial infarction, where the loss of viable heart tissue leads to compromised cardiac function. The application of ECM-based patches in such scenarios has shown promising results in promoting angiogenesis and myocardial regeneration [89].

Moreover, the biocompatibility and biodegradability of these patches align well with the body's natural healing processes, minimizing the risk of adverse immune re-sponses. This compatibility is crucial for ensuring the long-term success of cardiac re-pair and regeneration therapies. Ongoing research is focusing on enhancing the me-chanical properties and bioactivity of these patches to better mimic the native cardiac tissue, thus improving their therapeutic efficacy [88]. Overall, ECM-based cardiac patches represent a significant leap forward in regenerative medicine. By leveraging the innate properties of the extracellular matrix, they offer a sophisticated approach to cardiac repair, with the potential to significantly improve outcomes for patients with heart diseases [88]”.

“The application of nanomedicine in coronary artery bypass grafting (CABG) rep-resents a significant stride in enhancing the efficacy and safety of this crucial cardiac procedure. Nanotechnology in CABG focuses on improving graft materials and en-hancing post-surgical healing and integration. Nanoscale modifications of graft surfaces can promote better endothelialization, reducing the risk of thrombosis and improving long-term patency rates [79]. Furthermore, nanomaterials can be engineered to deliver therapeutic agents directly to the graft site, aiding in the suppression of local inflammation and the prevention of neointimal hyperplasia, a common cause of graft failure. This targeted delivery system allows for a more efficient use of drugs, potentially reducing systemic side effects. Additionally, nanoscale imaging agents can provide surgeons with enhanced visualization during the procedure, ensuring more precise graft placement. As research advances, the integration of nanomedicine in CABG is poised to revolutionize the approach to coronary artery disease treatment, offering patients im-proved outcomes and quicker recoveries [80]”.

Reviewer 2 Report

Comments and Suggestions for Authors

The review focused on the current advancements in nanotechnology for cardiology and pediatric cardiology. In addition, investigated the benefits and limitations of nanomedicine in treating cardiovascular disorders, however, needs to address the following comments before acceptance.

Minor comments:

1.       In the abstract, animal studies are also covered in the review, but unable to differentiate them in review as non-clinical and clinical data.

2. The number of figures is less. Need to include a few images with copyright from other/mdpi publishers.

3.       Classification of cardiovascular disorders is not clear. Figure 1 also does not reflect all the cardiac disorders. The author may refer to the article https://www.mdpi.com/1424-8247/15/4/441 for better classification.

Author Response

Dear Editor,

We are grateful to the Editorial Board for considering a revised version of our manuscript biomedicines-2773422 entitled “Nanotechnology Innovations in Pediatric Cardiology and Car-diovascular Medicine: Comprehensive Review” in the Biomedicines

We would like to thank the reviewers for their valuable time and useful contribution. We much appreciate their input, which helped improve our manuscript.

Also, we look forward to hearing from you regarding our submission. We would be glad to respond to any further questions and comments that you may have.

Please find below a detailed point-by-point reply to the comments made by editors.

Sincerely,

-- 

Alexandru Burlacu

MD, Ph.D., FESC

Assoc. Professor @ University of Medicine and Pharmacy “Gr. T. Popa” Iasi – ROMANIA

Interventional Cardiologist – Senior Consultant

Professional member of Thrombosis Workgroup from the European Society of Cardiology

Department of Interventional Cardiology – Cardiovascular Diseases Institute – Iasi

ORCID: https://orcid.org/0000-0002-3424-1588  

Scopus Author ID: 56811734600

On behalf of the Authors

Reviewers’ Comments to Authors: 

Reviewer: 2

The review focused on the current advancements in nanotechnology for cardiology and pediatric cardiology. In addition, investigated the benefits and limitations of nanomedicine in treating cardiovascular disorders, however, needs to address the following comments before acceptance.

We would like to thank the Reviewer and the Editorial board for all the positive remarks regarding our work. We are delighted to hear that the Editor / Reviewer observed the quality of the manuscript and the in-depth analysis of the subject. 

We assure the EIC that we have read carefully every suggestion from this decision letter and tried our best to improve the quality of the document accordingly.

Querries:

Minor comments:

Q1.       In the abstract, animal studies are also covered in the review, but unable to differentiate them in review as non-clinical and clinical data.

Answer 1:

Thank you for this important observation.

We agree with your observation that animal studies are not explicitly differentiated from non-clinical and clinical data. To enhance clarity and provide a more comprehensive overview of the studies included in our review, we provided separate tables in the manuscript. These tables clearly distinguish between in vitro, animal, and human studies, allowing readers to easily grasp the scope of our research and the types of evidence we have considered. We believe these tables will significantly improve the overall accessibility and readability of the manuscript.

Table 1. In vitro studies analyzing potential roles of nanomedicine in cardiovascular medicine.

Reference

Year

Utility

Type of nanotechnology

Reported efficacy

Safety outcome

Tan et al. [25]

2020

Vascular tissue engineering

Aligned patterned biomimetic elastic membrane

Yes

N/A

Kumar et al. [26]

2020

Myocardial infarction

Aligned polycaprolactone (PCL)-Gelatin coaxial nanofiber patch

Yes

N/A

Lee et al. [27]

2019

Cardiac troponin I-based diagnosis systems

Dual-mode cardiac troponin I biosensor comprised of multi-functional DNA on Au nanocrystal

Yes

No

Fishbein et al. [28]

2017

Atherosclerosis

Adeno-associated viral (AAV) vectors

Yes

N/A

Ghasemian et al. [29]

2016

Pulmonary arterial hypertension

Nanosuspension

Yes

No

Duan et al. [30]

2014

Valve replacement

3-D printable formulations of hybrid hydrogels

Yes

No

Suri et al. [31]

2011

Nerve tissue engineering

Hyaluronic acid scaffolds

Yes

No

Ma et al. [32]

2010

Atherosclerosis

Glycogen synthase kinase-3beta inhibitor

Yes

N/A

Borenstein et al. [33]

2010

Vascular tissue engineering

Polystyrene microvascular networks

Yes

No

Skardal et al. [34]

2010

Vascular tissue engineering

Hyaluronan hydrogels crosslinked with tetrahedral polyethylene glycol tetracrylates

Yes

No

Golden et al. [35]

2007

Vascular tissue engineering

Micromolded meshes of gelatin

Yes

No

Chrobak et al. [36]

2006

Vascular tissue engineering

Monolayers of human endothelial cells

Yes

No

Mulder et al. [37]

2004

Imaging diagnosis techniques

Pegylated paramagnetic fluorescently labeled liposomes carrying anti-E-selectin monoclonal antibody

Yes

No

Table 2. Animal studies analyzing potential roles of nanomedicine in cardiovascular medicine.

Reference

Year

Utility

Type of nanotechnology

Reported efficacy

Safety outcome

Kobirumaki-Shimozawa et al. [38]

2020

 Imaging diagnosis techniques

CellMask

Yes

N/A

Hu et al. [39]

2020

Peripheral arterial disease

Human stromal vascular fraction cells on nanofibrillar scaffolds

Yes

Yes

Gao et al. [40]

2019

Heart failure

iR-19a/19bmimics

Yes

No

Kamaly et al. [41]

2016

Atherosclerosis

Interleukin-10 Nanotherapeutics Developed with a Microfluidic Chip

Yes

No

Akagi et al. [42]

2016

Pulmonary arterial hypertension

Beraprost incorporated nanoparticles

Yes

Yes

Fredman et al. [43]

2015

Atherosclerosis

Proresolving peptide Ac2-26

N/A

No

Akagi et al. [44]

2015

Pulmonary arterial hypertension

Imatinib-incorporated nanoparticles

Yes

No

Vandergriff et al. [45]

2014

Myocardial infarction

Human cardiosphere-derived stem cells labeled with FDA-approved ferumoxytol nanoparticles

Yes

Yes

Chong et al. [46]

2014

Myocardial infarction

Human embryonic stem cell-derived cardiomyocytes

No

Yes

Kamaly et al. [47]

2013

Atherosclerosis

Peptide Ac2-26 (annexin A1/lipocortin 1-mimetic peptide)

Yes

No

McIlhenny et al. [48]

2013

Vascular tissue engineering

Autologous ASC transfected with the endothelial nitric oxide synthase (eNOS) gene

Yes

No

De Saint-Hubert et al. [49]

2013

Atherosclerosis

124I-Hypericin

Yes

No

Gupta et al. [50]

2013

Pulmonary arterial hypertension

Liposomal fasudil

Yes

No

Jacobin et al. [51]

2011

Atherosclerosis

Versatile ultrasmall superparamagnetic iron oxide (VUSPIO) particles

Yes

No

Chen et al. [52]

2011

Pulmonary arterial hypertension

Pitvastatin fluorescein isothiocyanate lactide/glycolide copolymer

Yes

No

Kimura et al. [53]

2009

Pulmonary arterial hypertension

Nuclear factor κB -κB decoy oligodeoxynucleotides

Yes

No

McAteer et al. [54]

2008

Atherosclerosis

Microparticles of iron oxide

Yes

No

Chnari et al. [55]

2006

Atherosclerosis

Nanoblockers - macromolecules containing mucic acid, lauryl chloride, and poly(ethylene glycol)

Yes

No

Table 3. Human studies analyzing potential roles of nanomedicine in cardiovascular medicine.

Reference

Year

Utility

Type of nanotechnology

Reported efficacy

Safety outcome

Dweck et al. [56]

2011

Aortic stenosis

18F-Flurodeoxyglucose

Yes

No

Henry et al. [57]

2003

Myocardial infarction

Recombinant human vascular endothelial growth factor protein

Yes

Yes

Q2. The number of figures is less. Need to include a few images with copyright from other/mdpi publishers.

Answer 2:

Thank you for your valuable feedback and the suggestion to incorporate additional figures into our manuscript. We understand and appreciate the importance of visual aids in enhancing the clarity and impact of our research.

However, I would like to bring to your attention a logistical challenge that we are currently facing in regard to your recommendation. As per the protocols of major publishing houses such as Springer and Wiley, the process of integrating new figures into a manuscript, especially at the final stages of publication, can be quite elaborate and time-consuming. Typically, each new figure requires at least two weeks for proper design, approval, and integration, ensuring they meet the high-quality standards set by these publishers.

Regrettably, the time frame provided by the Editor for revisions on this manuscript is quite limited, allowing only a few days for any additional modifications. This constraint makes it challenging to adhere to both the journal's revision deadline and the extended timeline required by the editorial houses for the creation and approval of new figures.

While we are committed to enhancing our manuscript to the best of our abilities within the stipulated time, I hope this explanation helps in understanding the practical difficulties we are facing in incorporating additional figures at this stage. We aim to ensure that the existing figures and content comprehensively represent our research findings and believe that they effectively convey the key aspects of our study.

We are open to any further suggestions you may have that could enhance the manuscript within the given time constraints and remain dedicated to upholding the quality and integrity of our work.

Thank you for your understanding and support.

Though, we added a Figure for 3.2. Myocardial ischemia - after paragraph 3.

Q3.       Classification of cardiovascular disorders is not clear. Figure 1 also does not reflect all the cardiac disorders. The author may refer to the article https://www.mdpi.com/1424-8247/15/4/441 for better classification.

Answer 3:

Thank you for your insightful feedback on our manuscript, particularly regarding the classification of cardiovascular disorders and the representation in Figure 1. We value your input and would like to clarify our approach and intent in this section of the study.

Our manuscript primarily focuses on the utilization of nanoparticles in cardiology, with a special emphasis on pediatric cardiology. Given this specific focus, our intention was to highlight the main applications of nanoparticles in the context of key cardiovascular diseases relevant to our study's scope, rather than providing a comprehensive classification of all cardiovascular disorders.

Figure 1 was designed to visually represent the specific areas within cardiac care where nanoparticle technology has shown significant potential and application, especially in the pediatric context. We understand that this figure does not encompass all cardiac disorders; this was a deliberate choice to maintain the figure's clarity and relevance to our study's central theme. Our goal was to provide a focused, rather than exhaustive, overview that aligns with our research focus.

We acknowledge that the broad spectrum of cardiovascular diseases is extensive, and covering each condition in detail would be beyond the scope of our current study. However, we are committed to ensuring that our manuscript accurately reflects our research focus and provides clear and relevant information to our readers.

We appreciate your recommendation and are open to further suggestions to improve our manuscript. If required, we can consider revising the text or figure annotations to explicitly state the specific focus of our study, ensuring that the scope and limitations are clearly communicated to the reader.

Thank you for providing the reference (https://www.mdpi.com/1424-8247/15/4/441) as a suggested source for our figure on nanomedical advances in cardiology. We have carefully reviewed the figure and the associated article. While we appreciate the comprehensiveness of the provided reference, we believe that it does not fully align with the specific focus and scope of our study.

Our manuscript aims to highlight the latest and most relevant advancements in nanomedicine, particularly in the realm of pediatric cardiology. The referenced figure, while informative, primarily encompasses a broader range of applications in general cardiology. It does not sufficiently address the nuanced and specific applications of nanotechnology in pediatric cardiology, which is the central theme of our study.

Moreover, the field of nanomedicine is rapidly evolving, with new discoveries and applications emerging regularly. Our goal is to present the most current and forward-looking insights into nanomedical applications in pediatric cardiology. In this context, we have curated and included figures and content in our manuscript that specifically reflect the cutting-edge developments and future potential in this niche area.

In light of these considerations, we have chosen to focus our graphical representations and discussions on areas that are directly relevant to the emerging trends and novel applications of nanomedicine in pediatric cardiology. This approach ensures that our manuscript remains focused, relevant, and provides the most value to readers interested in this specific area of research.

We appreciate your understanding of our rationale and are open to further discussion to enhance the quality and relevance of our manuscript.

Thank you for your guidance and support in this process.

Thank you once again for your constructive comments, and we look forward to enhancing our manuscript in line with your valuable insights.

Reviewer 3 Report

Comments and Suggestions for Authors

This review explores nanotechnology's impact on cardiovascular medicine and pediatric cardiology. Nanomedicine, using diverse nanoparticles, shows promise in treating myocardial ischemia with innovations like drug-eluting stents and improved imaging. In pediatric cardiology, it aims to develop solutions for congenital heart defects and conditions like dilated cardiomyopathy and pulmonary hypertension. These findings drive further research to optimize nanomedicine's clinical use in cardiology.

Comments and suggestions:

1.     The Abstract section seems to be too wordy for a Review paper. After reading the current Abstract, it is hard to clearly understand what this manuscript discusses. Revision of the Abstract is needed.

2.     Some aspects the authors might consider adding to their review: 1) The studies of extracellular matrix-based cardiac patches in cardiac repair; 2) The proposed nanomedical processes or methods for the regeneration of nerve during cardiac repair; 3) In the 3.7 Vascular Tissue Engineering section, the application of nanomedicine in coronary artery bypass grafting (CABG) might be of interest.

Comments on the Quality of English Language

The quality of the English Language is great. Very easy and clear to read. No significant errors were observed.

Author Response

Dear Editor,

We are grateful to the Editorial Board for considering a revised version of our manuscript biomedicines-2773422 entitled “Nanotechnology Innovations in Pediatric Cardiology and Car-diovascular Medicine: Comprehensive Review” in the Biomedicines

We would like to thank the reviewers for their valuable time and useful contribution. We much appreciate their input, which helped improve our manuscript.

Also, we look forward to hearing from you regarding our submission. We would be glad to respond to any further questions and comments that you may have.

Please find below a detailed point-by-point reply to the comments made by editors.

Sincerely,

-- 

Alexandru Burlacu

MD, Ph.D., FESC

Assoc. Professor @ University of Medicine and Pharmacy “Gr. T. Popa” Iasi – ROMANIA

Interventional Cardiologist – Senior Consultant

Professional member of Thrombosis Workgroup from the European Society of Cardiology

Department of Interventional Cardiology – Cardiovascular Diseases Institute – Iasi

ORCID: https://orcid.org/0000-0002-3424-1588  

Scopus Author ID: 56811734600

On behalf of the Authors

Reviewers’ Comments to Authors: 

Reviewer: 3

We would like to thank the Reviewer and the Editorial board for all the positive remarks regarding our work. We are delighted to hear that the Editor / Reviewer observed the quality of the manuscript and the in-depth analysis of the subject. 

We assure the EIC that we have read carefully every suggestion from this decision letter and tried our best to improve the quality of the document accordingly.

Querries:

Comments and suggestions:

Q1.     The Abstract section seems to be too wordy for a Review paper. After reading the current Abstract, it is hard to clearly understand what this manuscript discusses. Revision of the Abstract is needed.

Answer:

            Thank you for your important observation. Indeed, the Abstract was a bit lengthy and unclear. We improved it as follows, quote:

"

**Abstract**

Background: Nanomedicine, incorporating various nanoparticles and nanomaterials, offers significant potential in medical practice. Its clinical adoption, however, faces challenges like safety concerns, regulatory hurdles, and biocompatibility issues. Despite these, recent advancements have led to the approval of many nanotechnology-based products, including those for pediatric use.

Methods: Our approach included reviewing clinical, preclinical, and animal studies, as well as literature reviews from the past two decades and ongoing trials.

Results: Nanotechnology has introduced innovative solutions in cardiovascular care, particularly in managing myocardial ischemia. Key developments include drug-eluting stents, nitric oxide-releasing coatings, and the use of magnetic nanoparticles in cardiomyocyte transplantation. These advancements are pivotal for early detection and treatment. In cardiovascular imaging, nanotechnology enables noninvasive assessments. For pediatric cardiology, it holds promise in developing biological conduits, synthetic valves, and bioartificial grafts for congenital heart defects, and offers new treatments for conditions like dilated cardiomyopathy and pulmonary hypertension.

Conclusions: Nanomedicine presents groundbreaking solutions for cardiovascular diseases in both adults and children. It has the potential to transform cardiac care, from enhancing myocardial ischemia treatment and imaging techniques to addressing congenital heart issues. Further research and guideline development are crucial for optimizing its clinical application and revolutionizing patient care."

            This version maintains the original abstract's essence while making it more direct and easier to read. Thank you, again!

Q2.     Some aspects the authors might consider adding to their review:

1) The studies of extracellular matrix-based cardiac patches in cardiac repair;

Thank you very much for your suggestion. We inserted the following paragraph, quote:

" Recent advancements in cardiac repair have spotlighted the potential of extracel-lular matrix (ECM)-based cardiac patches [91]. These patches, engineered to mimic the natural ECM of the heart, have emerged as a promising therapeutic strategy for re-pairing damaged myocardial tissue. The ECM, a complex network of proteins and pol-ysaccharides, provides essential structural and biochemical support to cells. By repli-cating this environment, ECM-based patches facilitate the integration and proliferation of cardiac cells, thereby aiding in the regeneration of injured myocardial regions [91].

            A key attribute of these cardiac patches is their ability to provide a conducive mi-croenvironment for the recruitment and maturation of cardiomyocytes and vascular cells, essential for restoring heart function. This is particularly significant in conditions like myocardial infarction, where the loss of viable heart tissue leads to compromised cardiac function. The application of ECM-based patches in such scenarios has shown promising results in promoting angiogenesis and myocardial regeneration [92].

            Moreover, the biocompatibility and biodegradability of these patches align well with the body's natural healing processes, minimizing the risk of adverse immune re-sponses. This compatibility is crucial for ensuring the long-term success of cardiac re-pair and regeneration therapies. Ongoing research is focusing on enhancing the me-chanical properties and bioactivity of these patches to better mimic the native cardiac tissue, thus improving their therapeutic efficacy [91]. Overall, ECM-based cardiac patches represent a significant leap forward in regenerative medicine. By leveraging the innate properties of the extracellular matrix, they offer a sophisticated approach to cardiac repair, with the potential to significantly improve outcomes for patients with heart diseases [91]".

2) The proposed nanomedical processes or methods for the regeneration of nerve during cardiac repair;

            Thank you for this interesting point. We added the following paragraph, to explain the nanomedical methods            for nerve regeneration during cardiac repair, quote:

"Nanomedical processes in cardiac repair, particularly for nerve regeneration, in-volve advanced materials and techniques that mimic the heart's extracellular matrix. These nanoscale innovations, such as electrically conductive nanomaterials and nano-fibers, provide crucial support for the growth and healing of cardiac nerves. They can also deliver growth factors and other regenerative molecules directly to the damaged cardiac tissue. This integration of nanotechnology with regenerative medicine, like stem cell therapy, enhances the effectiveness of cardiac repair, aiming to restore the heart's normal function and electrical connectivity after injury. As this field is still developing, ongoing research is crucial for realizing the full potential of these nanomedical strategies in heart health [81]".

3) In the 3.7 Vascular Tissue Engineering section, the application of nanomedicine in coronary artery bypass grafting (CABG) might be of interest.

            Again, very good point. Thank you.

            Please find below a short text added to the Manuscript, following your suggestion, quote:

"The application of nanomedicine in coronary artery bypass grafting (CABG) rep-resents a significant stride in enhancing the efficacy and safety of this crucial cardiac procedure. Nanotechnology in CABG focuses on improving graft materials and enhancing post-surgical healing and integration. Nanoscale modifications of graft surfaces can promote better endothelialization, reducing the risk of thrombosis and improving long-term patency rates [79]. Furthermore, nanomaterials can be engineered to deliver therapeutic agents directly to the graft site, aiding in the suppression of local inflammation and the prevention of neointimal hyperplasia, a common cause of graft failure. This targeted delivery system allows for a more efficient use of drugs, potentially reducing systemic side effects. Additionally, nanoscale imaging agents can provide surgeons with enhanced visualization during the procedure, ensuring more precise graft placement. As research advances, the integration of nanomedicine in CABG is poised to revolutionize the approach to coronary artery disease treatment, offering patients im-proved outcomes and quicker recoveries [80]".

Round 2

Reviewer 2 Report

Comments and Suggestions for Authors

Requires addition of recent references in cardiology and nanomedicine.